# Crystal structure of dopamine receptor D4 bound to the subtype selective ligand, L745870

Ye Zhou[1,2], Can Cao[1], Lingli He[1], Xianping Wang[1], Xuejun Cai Zhang[1,2]*

[1]National Laboratory of Biomacromolecules, CAS Center for Excellence in Biomacromolecules, Institute of Biophysics, Chinese Academy of Sciences, Beijing, China; [2]University of Chinese Academy of Sciences, Beijing, China

**Abstract** Multiple subtypes of dopamine receptors within the GPCR superfamily regulate neurological processes through various downstream signaling pathways. A crucial question about the dopamine receptor family is what structural features determine the subtype-selectivity of potential drugs. Here, we report the 3.5-angstrom crystal structure of mouse dopamine receptor D4 (DRD4) complexed with a subtype-selective antagonist, L745870. Our structure reveals a secondary binding pocket extended from the orthosteric ligand-binding pocket to a DRD4-specific crevice located between transmembrane helices 2 and 3. Additional mutagenesis studies suggest that the antagonist L745870 prevents DRD4 activation by blocking the relative movement between transmembrane helices 2 and 3. These results expand our knowledge of the molecular basis for the physiological functions of DRD4 and assist new drug design.

## Introduction

Dopamine and its receptors play fundamental roles in controlling neurological processes including motor control, cognition, learning, and emotions (*Beaulieu and Gainetdinov, 2011*). Dopaminergic dysfunction is associated with the pathophysiology of various disorders such as schizophrenia, mood disorders, attention-deficit disorder, Tourette's syndrome, substance dependency, and Parkinson's disease (*Girault and Greengard, 2004*). The dopamine receptor family, which belongs to the G-protein-coupled receptor (GPCR) superfamily, can be divided into two subfamilies, namely the D1-like subfamily (including dopamine receptor D1 (DRD1) and DRD5), and the D2-like subfamily (including DRD2, DRD3, and DRD4). These subfamilies probably originated from two independent acquirements of the ability of dopamine binding during the evolution of biogenic amine receptors (*Opazo et al., 2018*). The D1- and D2-like subfamilies are coupled to different G proteins, activating distinct downstream signaling pathways (*Neve et al., 2004*). In particular, DRD1 and DRD5 are found exclusively in postsynaptic sites of dopamine-receptive neurons. After binding dopamine, they activate $G\alpha_{s/olf}$ proteins and upregulate the adenylyl cyclase activity in the cell. In comparison, DRD2, DRD3, and DRD4 are expressed both presynaptically on dopaminergic neurons and postsynaptically on dopamine target cells. They activate $G\alpha_{i/o}$ proteins and thus downregulate the adenylyl cyclase activity (*Sokoloff, 2006*; *Rondou et al., 2010*).

Recent advances in GPCR structure biology make it possible to solve the mystery of dopamine receptors and their signaling pathways. To date, the structures of all three types of D2-like receptors have been reported: In 2010, the crystal structure of human DRD3 was solved (*Chien et al., 2010*), followed by crystal structures of human DRD4 (*Wang et al., 2017*) and DRD2 (*Wang et al., 2018*). In agreement with their high degree of sequence homology, ligand-binding pockets in these D2-like receptors share many structural features in addition to their conserved overall folding. These structural studies expand our understanding of the distinctive selectivity spectrum of the binding pockets

*For correspondence:
zhangc@ibp.ac.cn

Competing interests: The authors declare that no competing interests exist.

of dopamine receptors toward different ligands including those belonging to the same chemical category but containing subtle modifications.

DRD4 is characterized by its more selective expression pattern in the brain (such as the frontal cortex and amygdala; *Murray et al., 1995*), and thus exhibits promising pharmacological properties. Their anatomical localization in the prefrontal cortex, in contrast to little or no expression in the basal ganglia, strongly indicates roles of DRD4 receptors in cognition and emotions. Although dopamine is the most potent endogenous ligand known to activate DRD4, the receptor is also readily activated by epinephrine and norepinephrine (*Newman-Tancredi et al., 1997*). In addition, clozapine binds to DRD4 with an affinity 10-fold higher than that to other dopaminergic receptors (*Boeckler et al., 2004*; *Wilson et al., 1998*). To date, the highest degree of selectivity for dopaminergic receptors having been achieved by pharmaceutics are certain DRD4-selective antagonists, exhibiting more than 1000-fold higher affinity towards D4 receptors compared to other dopaminergic receptors (*Rondou et al., 2010*). One well-characterized selective DRD4-antagonist with high affinity ($K_i$ ~0.43 nM) is a 1,4-disubstituted aromatic piperidine/piperazine (1,4-DAP) compound named L745870 (*Patel et al., 1997*). L745870 has been extensively analyzed for its pharmacology in various behavioral tests (in rodents) and clinical trials for treating schizophrenia (*Patel et al., 1997*; *Bristow et al., 1997*).

To facilitate future pharmacological studies and possible improvement of related compounds, we determined the 3.5 Å resolution crystal structure of mouse DRD4 (66% identical to the human DRD4 in the whole amino acid sequence and 89% identical in the transmembrane (TM) region) in complex with L745870. This complex structure shows the existence of an extended ligand-binding pocket specific for DRD4s and improved our understanding of the structural bases of subfamily-selectivity of potential antagonists.

## Results

### Overall structures

To crystallize the mouse DRD4 (simply referred to as DRD4 hereafter unless otherwise specified), we used protein engineering (*Chun et al., 2012*; *Cao et al., 2018*) in combination with lipidic cubic phase (LCP) methods (*Caffrey, 2009*). Recombinant proteins were purified and crystallized in the presence of the subtype-selective antagonist, L745870, to stabilize the receptor (*Figure 1—figure supplement 1*). To aid crystallization, we replaced the long, presumably disordered, intracellular loop 3 (ICL3, that is residues A220$^{5.70}$–A302$^{6.24}$) of the receptor with a thermo-stable variant of apocytochrome b$_{562}$ (BRIL) (*Chun et al., 2012*). To improve the diffraction quality of the DRD4 crystal, we truncated the N-terminal 22 amino acid residues, which fail to adopt an ordered secondary structure according to bioinformatic predictions. This N-terminal peptide was replaced by a FLAG-tag without interfering with the overall structure. To the contrary, the tag appeared to facilitate the crystallization process. The crystallized recombinant DRD4 protein contained four additional point mutations, F121$^{3.41}$W, P201$^{5.52}$I, P317$^{6.38}$A, and C181$^{45.51}$R (superscript numerals refer to the Ballesteros-Weinstein numbering system; *Ballesteros and Weinstein, 1995*). The first mutation is based on a previously reported stabilizing mutation in class-A GPCRs (*Roth et al., 2008*) (*Figure 1—figure supplement 1a*). The remaining three mutations were designed based on sequence comparison of the D2-like subfamily (*Figure 1—figure supplement 2*) and were predicted not to impair the native binding ability of the receptor. In particular, P201$^{5.52}$I and P317$^{6.38}$A were found to be necessary for proper crystallization, while C181$^{45.51}$R improved the diffraction quality. The combination of these modifications (including the BRIL fusion) increased the melting temperature (Tm) of the protein variant by 13°C compared to the wild-type protein (WT, without the ICL3 substitution), and the presence of L745870 further increased the Tm by 20°C (*Figure 1—figure supplement 1a*). The DRD4 crystal structure was refined at 3.5 Å resolution (*Supplementary file 1*). Furthermore, a recently reported technique utilizes a cpGFP (circular permuted GFP)-fusion dopamine receptor variant as an in vivo dopamine-binding reporter (*Patriarchi et al., 2018*; *Sun et al., 2018*). Based on this method, we confirmed that the mutation variant used for structural determination (with BRIL replaced by cpGFP) maintained significant ability to response to both agonists and antagonists (*Figure 2—figure supplement 1a*).

The overall structure of the TM domain of mouse DRD4 was similar to the previously reported ground-state structures of human DRD4 and DRD3 (*Chien et al., 2010*; *Wang et al., 2017*). Our DRD4 crystal possessed a typical type-I crystal packing of membrane proteins, with the TM regions forming continuous layers. Moreover, DRD4 formed a non-symmetrical dimer in each crystallography asymmetric unit. The two protomers are denoted as Mol-A and Mol-B, and are related to each other by a 44° rotation about an axis perpendicular to the membrane plane in addition to a non-crystallographic translation (*Figure 1a,b*). TMs 5 and 6 of the receptor form continuous helices with a helix pair from the helix bundle of the BRIL fusion. This intact helix structure seems to provide more rigid packing which may explain why the P201$^{5.52}$I and P307$^{6.38}$A mutations facilitated crystallization. Structures of the FLAG-tag and N-terminal residues 23–30 failed to be resolved, presumably due to their flexibility or multiple conformations. Well-defined electron densities were observed for ICL1, extracellular loop-1 (ECL1), and ECL3. In contrast, electron density of ECL2 in Mol-B broke near the signature C105$^{3.25}$-C180$^{45.50}$ disulfide bond, and residues between the C-terminal end of TM4 and the conserved C180$^{45.50}$ were unsolvable from the density. In comparison, the ECL2 of Mol-A showed a clear electron density at the C-terminal end of TM4, but became disordered near the N-terminal end of TM5. This stability difference between the two ECL2 loops is likely to be due to distinct crystal packing interactions (*Figure 1—figure supplements 3* and *4*). Based on the quality of the electron density map as well as the distribution of B-factors (*Figure 1—figure supplement 5*), the most stable regions of the entire dimer were the dimer interface and BRIL helices fused with

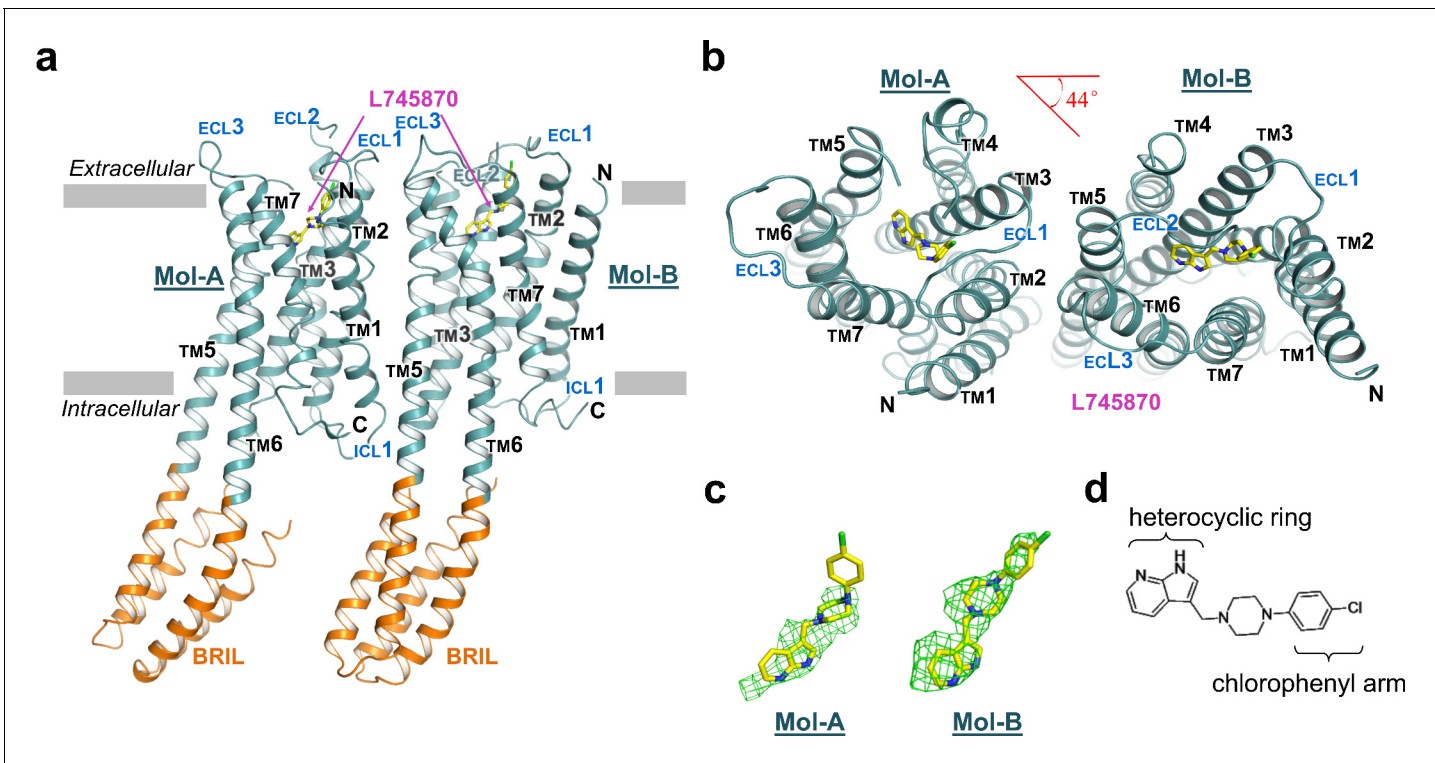

**Figure 1.** The overall structure of mouse DRD4 in complex with L745870. (a) Ribbon representation of mouse DRD4 (TM domain presented in cyan, ligand in yellow, and BRIL fusion in orange), as viewed parallel to the membrane with approximate membrane boundaries indicated with gray lines. (b) Top view of the mouse DRD4 structure. Rotation angle between Mol-A and Mol-B calculated with PyMOL is shown as a red angle sign. (c) The L745870 antagonist is shown in stick representation with carbon, nitrogen and chloride atoms shown in yellow, blue, and green, respectively. Fo−Fc omit density map for L745870 is contoured at 2.5 σ. (Also see *Figure 1—figure supplement 5*) (d) Chemical structure of L745870.

The online version of this article includes the following figure supplement(s) for figure 1:

**Figure supplement 1.** Profiles of DRD4 in different constructs or under different conditions.

**Figure supplement 2.** Protein sequence alignment of D2-like family members.

**Figure supplement 3.** Crystal packing of the mouse DRD4.

**Figure supplement 4.** Interactions of different asymmetric units in mouse DRD4 crystal packing.

**Figure supplement 5.** The B-factors distribution and representative electron density.

TMs 5 and 6 in Mol-B. Thus, the dimer appeared to serve as a structural unit of crystal packing. In comparison, the BRIL molecule fused with Mol-A exhibited a higher flexibility. The C-termini of the both protein molecules were also disordered, with only a short segment of helix-8 being resolved in each protomer.

## Ligand-binding pocket

Clear electron density was observed in the ligand-binding pocket in Mol-B; in comparison, the electron-density omit map was murky at the corresponding position in Mol-A (*Figure 1c*). Nevertheless, including L745870 in Mol-A in a conformation similar to that in Mol-B during refinement successfully reduced the R-factors, albeit only slightly. Because of the limited resolution of our crystal structure, we do not further speculate the biological significance of the differential binding of L745870 in the non-symmetrical dimer (*Figure 2—figure supplement 2*). Here, we describe the binding pocket of DRD4 on the basis of L745870 bound to Mol-B. This antagonist was buried in the upper part of the TM core underneath the signature C105$^{3.25}$-C180$^{45.50}$ disulfide bond (*Figure 2a,b*). The binding pocket was formed by residues from TMs 2, 3, 5, 6, and 7. Furthermore, L745870 is a typical 1,4-DAP compound (*Figure 1d*), which presents a common characteristic of many high-selectivity ligands of D4 receptors (*Oak et al., 2000*). A proton-titratable amine group exists in the core piperazine on the arm side of the heterocyclic ring (*Kortagere et al., 2004*), and formed a hydrogen bond (2.9 Å) with the acidic residue, D112$^{3.32}$, conserved in all members of the aminergic receptor family. Simultaneously, the carboxyl sidechain of D112$^{3.32}$ formed a hydrogen bond with the hydroxyl group of Y358$^{7.43}$, and this hydrogen bond is also conserved in almost all aminergic receptor structures (*Wang et al., 2017*; *Wang et al., 2013*; *Liu et al., 2013*; *Chung et al., 2011*; *Thal et al., 2016*). D112$^{3.32}$ serves as the counter ion for the protonated amine group in neurotransmitters such as dopamine, serotonin, and epinephrine (*van Rhee and Jacobson, 1996*).

## Orthosteric pocket

The heterocyclic ring of L745870 resides in the relatively hydrophobic orthosteric pocket, surrounded by V113$^{3.33}$, F330$^{6.51}$, and F331$^{6.52}$ from both sides, and by C116$^{3.36}$ and W327$^{6.48}$ from the bottom. In our cpGFP-based dopamine-activation assay, the F330$^{6.51}$Y mutation, which left the aromatic moiety unchanged, showed no detectable defect on the binding of dopamine or other ligands used (*Figure 2—figure supplement 1b*).

The well-conserved residues S191$^{5.42}$ and S192$^{5.43}$, which are postulated to function as hydrogen-bond acceptors for the hydroxyl groups of bioamines during dopamine activation (*Strader et al., 1989*; *Cox et al., 1992*), were found to be positioned away from L745870 in our complex structure, probably because of the lack of hydroxyl groups in the pyridine part of the ligand. Moreover, our cpGFP-based dopamine-activation assay (*Figure 2—figure supplement 1b*) showed that, to induce the fluorescence intensity change (*i.e.*, to activate the receptor), the S191$^{5.42}$A -containing variants required 100-fold higher concentrations of dopamine compare to the WT construct. In comparison, the S192$^{5.43}$A mutation slightly interfered with the dopamine-induced conformational change. In addition, the S195$^{5.46}$A mutation completely abolished the activating ability of dopamine. In agreement with previous reports (reviewed in *Michino et al., 2015*), our functional studies suggest that the three adjacent serine residues within the orthosteric pocket play crucial roles in receptor activation by dopamine.

## Extended pocket

The chlorophenyl arm of L745870 was observed to protrude into an extended pocket of our mouse DRD4 structure, which is also present in the human DRD4 structure (*Wang et al., 2017*). In particular, the ligand-binding pocket of DRD4 extended from the orthosteric pocket into a crevice between TM2 and TM3, rather than a crevice between TM2 and TM7 as observed in the D2 and D3 receptors (*Figure 2c,d*). This extended pocket was surrounded by residues L83$^{2.56}$, F88$^{2.61}$, S91$^{2.64}$, L108$^{3.28}$, and M109$^{3.29}$, with the signature C105$^{3.25}$-C180$^{45.50}$ disulfide bond and W98$^{23.50}$ (a conserved residue in the connecting loop between TMs 2 and 3) located on the top of the pocket. However, a few structural features around TM2 appear to limit further extension of the pocket. For instance, while providing hydrophobic interactions with the piperazine group of L745870, F88$^{2.61}$ obstructed further extension. Similarly, a hydrogen bond between E92$^{2.65}$ and S351$^{7.36}$ appeared to function as a lock

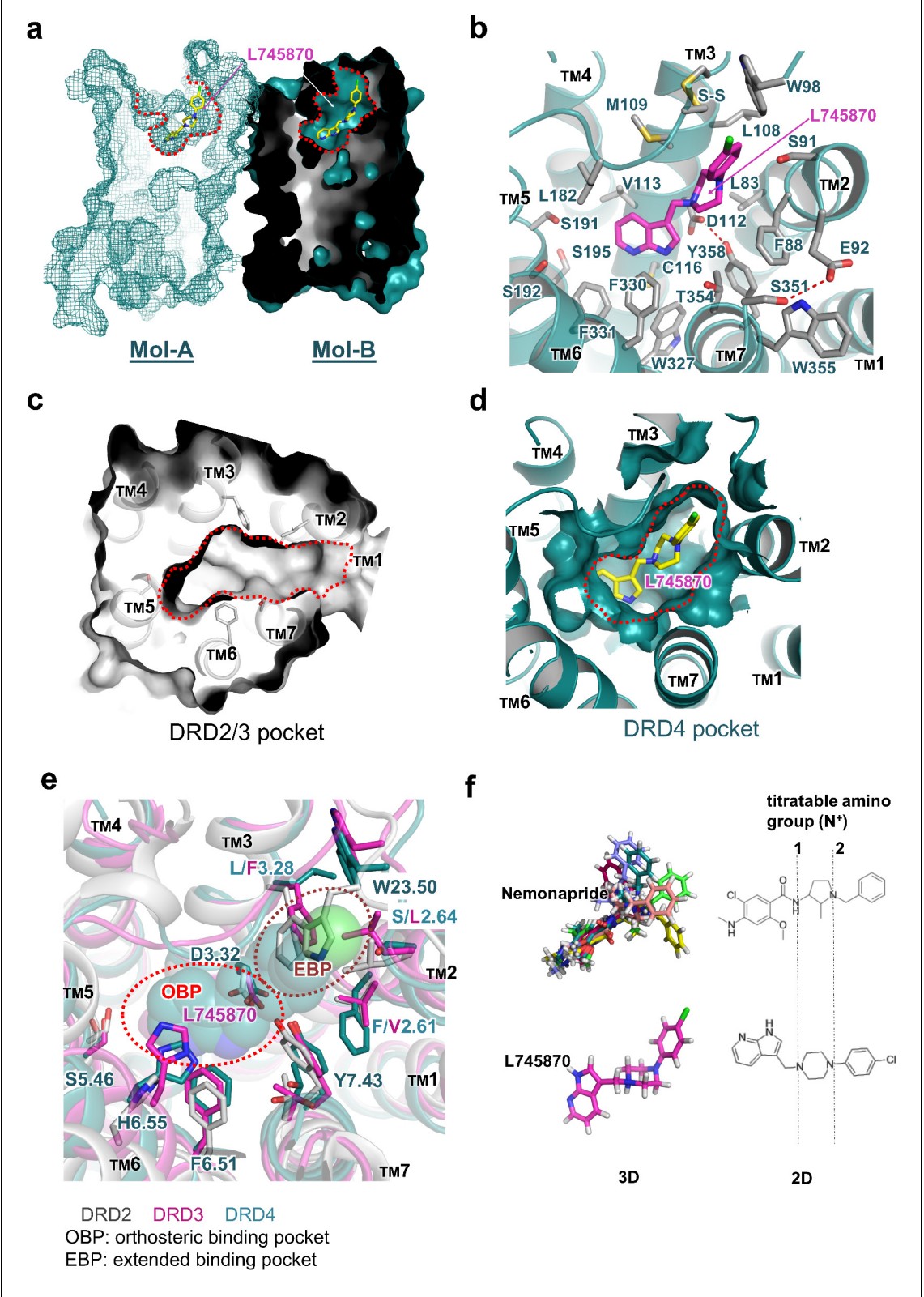

**Figure 2.** Molecular details of the L745870-binding site in DRD4. (a) Mesh (Mol-A) and surface (Mol-B) representation of mouse DRD4 as viewed parallel to the membrane plane, clipped to reveal L745870 (colored as in *Figure 1*). (b) Interactions between the residues confining the binding pocket and L745870 (shown in magenta), with potential polar interactions depicted as dashed red lines. (c, d) Top view of surface representation of binding pocket in DRD2/3 and DRD4, respectively. L745870 is colored as in *Figure 1*. (e) Structural differences in the extended binding pockets (EBPs) of DRD4 (cyan),

*Figure 2 continued on next page*

*Figure 2 continued*

DRD2 (gray), and DRD3 (magenta). L745870 is shown as a space-filling model with carbon, nitrogen, and chloride atoms shown in cyan, blue, and green, respectively. (f) Comparison of nemonapride and L745870 in 3D models as well as 2D structures. The protonatable amino groups are marked with dashed lines in the 2D structures. For nemonapride, protonation occurs both at positions 1 and 2, whereas for L745870 it occurs mainly at position 1. The 3D steric conformations were generated with Maestro using the ligand preparation mode.

The online version of this article includes the following figure supplement(s) for figure 2:

**Figure supplement 1.** Representative results from Ligand binding-induced fluorescence intensity change with DRD4-cpGFP based dopamine assay.
**Figure supplement 2.** Structural comparison of mouse and human DRD4 receptors.

restricting the extension (*Figure 2b*). In agreement with these structural observations, our cpGFP-based dopamine-activation assays showed that small-to-large mutations $S91^{2.64}L$ and $L108^{3.28}F$ dramatically reduced the inhibitory effects of L745870 against dopamine activation. Even more drastically, the $S91^{2.64}F$ mutation abolished the response of the receptor to dopamine (*Figure 2—figure supplement 1c*).

## Dimerization

Our crystal structure of DRD4 revealed a non-symmetrical interface between TMs 1–3 of Mol-A and TMs 5 and 6 of Mol-B. Following dimerization, a total of 1600 $Å^2$ 'solvent-accessible surface' was buried within the dimer. To our knowledge, such a non-symmetrical homodimer has not been described in crystal structures of GPCRs before. The dimerization was stabilized by hydrogen bonds from both extracellular and intracellular regions, as well as hydrophobic packing between the transmembrane domains (*Figure 3a*). In particular, a polar interaction between $S101^{A3.21}$ and $D186^{B5.37}$ (the superscript letter, A or B, denotes the host protomer of the corresponding residue) linked the extracellular sides of $TM3^A$ and $TM5^B$, whereas a hydrophobic interaction on the same side of $TM2^A$ and $TM6^B$ was formed through $Y90^{A2.63}$ and $L339^{B6.60}$, together with a disulfide bridge within ECL3 of Mol-B (*Figure 3b*). On the intracellular side, two hydrogen bonds (namely between the mainchain of $A57^{A1.58}$ and the sidechain of $T135^{B3.55}$, and between the sidechain of $S58^{A1.59}$ and the mainchain of $G213^{B5.64}$) and a salt-bridge bond between $E59^{A1.60}$ and $R216^{B5.67}$ stabilized the packing of $TM1^A$, $TM3^B$, and $TM5^B$ (*Figure 3c*). These interacting regions formed a stabilizing triangle, locking the dimer interface of aliphatic interactions of the TM region in a stable conformation. Results from our cross-linking experiment further suggest that dimerization of recombinant DRD4 molecules also occurs in solution (*Figure 3—figure supplement 1*).

## Discussion

In this study, we present the structure analysis on DRD4 complexed with its antagonist L745870. Comparison of our complex structure with known structures of DRD2, DRD3, and DRD4 reveals the structural basis of subtype-selectivity (*Figure 2e*). As suggested by previous reports (*Simpson et al., 1999*; *Schetz et al., 2000*), the subtype-specific determinants that contribute to the ligand selectivity within the D2-like subfamily are located near the extracellular ends of TMs 2 and 3. Receptors of this subfamily share the same set of residues in the orthosteric binding pocket to maintain dopamine binding ability. More importantly, less conserved residues in TMs 2 and 3 contribute micro-patches to the pocket that discriminates different substitutions of the ligands. For instance, in DRD2/3-antagonist complex structures, a smaller aliphatic sidechain at $V^{2.61}$ (compared to $F88^{2.61}$ in DRD4) allows ligands specific to the DRD2/3 subfamily to be oriented toward a crevice between TMs 1–2 and TM7 (*Chien et al., 2010*). In addition, previous experiments substituting amino acid residues at less conserved positions in the D4 receptor with corresponding residues from DRD2 or constructing D2/D4 and D4/D2 hybrids with the chimeric junction located near the N-terminal portion of TM3 showed to switch the pharmacological profiles of the variants. In particular, DRD4-selective ligands showed decreased binding affinity to DRD4 variants constructed in such a manner (*Kortagere et al., 2004*; *Shih et al., 1997*). Importantly, steric clashes will occur at positions of $S^{2.64}L$ and $L^{3.28}F$ in DRD2/3 or $W^{23.50}$ in DRD2, if L745870 is superimposed into their corresponding ligand-binding pockets (*Figure 2*).

From a pharmaceutical point of view, pluridimensional efficacy and adverse effects of a given drug result from its nonspecific interactions with unintended targets. Therefore, increasing the

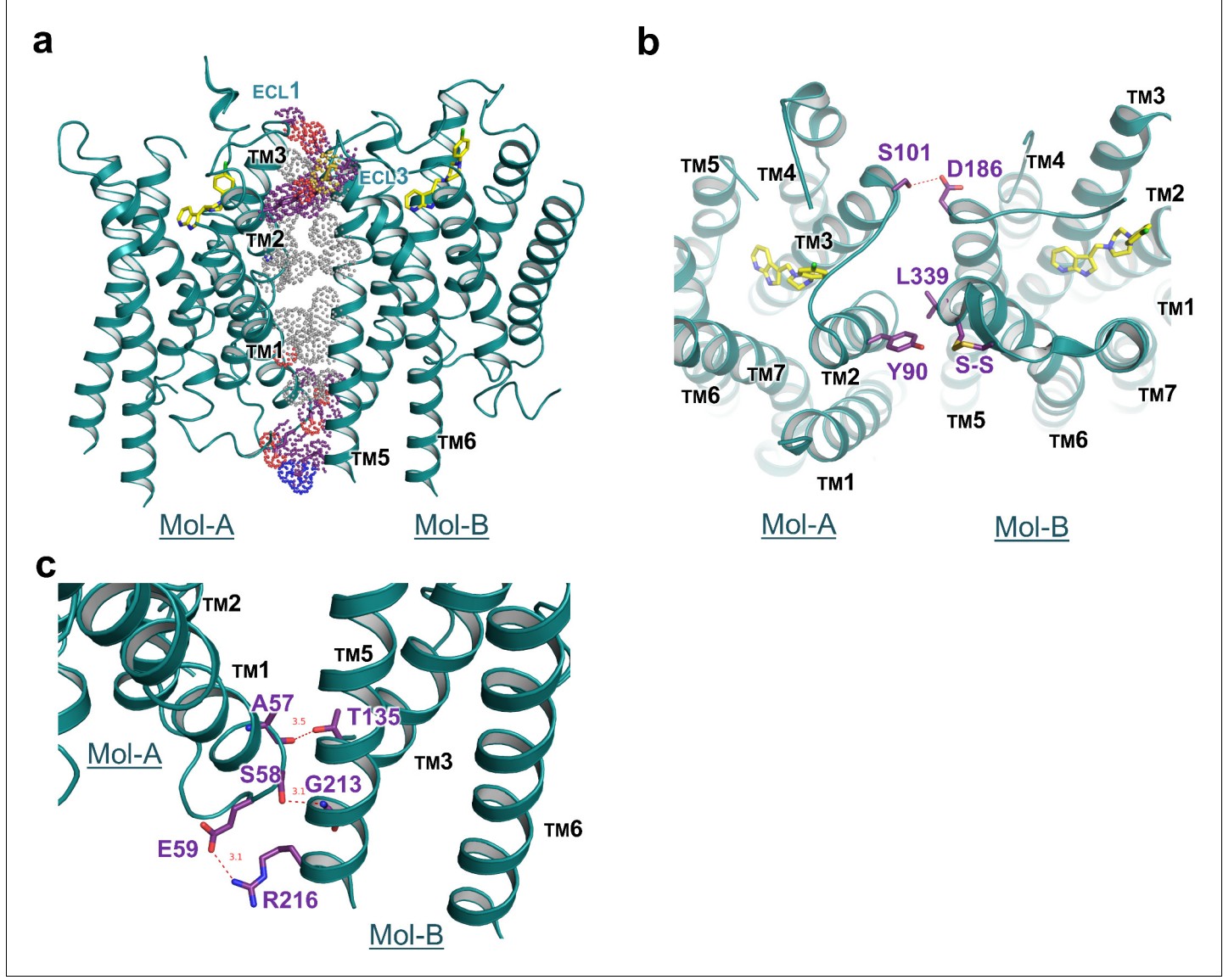

**Figure 3.** Interface details of mouse DRD4 in the asymmetric dimer. (a) Overview of the interface of the mouse DRD4 dimer. Dot representations of the interacting residues are shown for the transmembrane domain (gray) and exo-membrane regions (colored). (b, c) Extracellular and intracellular faces of the DRD4 dimer. Red dashed lines depict potential polar interactions.

The online version of this article includes the following figure supplement(s) for figure 3:

**Figure supplement 1.** DRD4 cross-linked with glutaraldehyde.

receptor-discriminating ability of lead compounds has long been a major focus of drug design and pharmaceutics. Ligands bound in the previously reported crystal structures of the D2, D3, and D4 receptors are risperidone, eticlopride, and nemonapride, respectively, all of which represent non-selective ligands. In particular, risperidone is known to bind to all D2-like receptors as well as other types of aminergic receptors (*Janssen et al., 1988*); eticlopride is a DRD2/D3-selective antagonist (*Jensen, 2015*); and nemonapride is both an antagonist of D2-like receptors and a potent agonist of the serotonin 1A (5-HT1A) receptor (*Seeman et al., 1993*). Our crystal structure of the DRD4-L745870 complex provides a critical model to decipher the subtype selectivity of ligands, a characteristic of the D2-like receptors. Most classic biogenic amine ligands of aminergic receptors (agonists or antagonists, including L745870) possess a protonated amine group that binds the conserved $D^{3.32}$ site (*Thal et al., 2016*; *Shimamura et al., 2011*; *Ring et al., 2013*; *Miller-Gallacher et al.,*

*2014*). This specific interaction partially explains why L745870 and dopamine compete for binding the DRD4. Furthermore, by reducing the moving freedom of TM3 required for full activation of class-A GPCRs (*Hulme, 2013*), the bulky chemical group of L745870 inserted into the TM2-TM3 crevice may provide a structure basis for its antagonist activity. Being consistent with this notion, the results of our mutational analysis showed that, when substituting $S^{2.64}$ in the TM2-TM3 cavity with a bulky sidechain (such as Leu or Phe), the dopamine-induced activity was decreased or totally abolished (*Figure 2* and *Figure 2—figure supplement 1*). To further elaborate the discrepancy between the chemical structures of L745870 and nemonapride and its relationship to the binding modes inside DRD4, we performed a molecular docking analysis using the program Schrodinger (*Friesner et al., 2006*). Under the same set of simulation conditions, a nemonapride molecule assumes up to 11 steric conformations with each containing two protonated sites; in contrast, two conformations are observed for L745870, and only one of them is able to form a hydrogen bond to $D^{3.32}$(*Figure 2f*). Upon anchoring the central proton-titratable amino group of the ligand to $D^{3.32}$, the more hydrophilic entity on one side occupies the dopamine-binding orthosteric pocket, while the more hydrophobic entity on the other side protrudes into the extended pocket. Torsion angles between the three parts appear to play important roles in determining the ligand-binding mode. These observations clearly explain why nemonapride can bind to both D4 and D2/3 receptors whereas L745870 possesses high selectivity toward the D4 receptor, thus suggesting a direction to improve the ligand potency for the D2-like subfamily.

GPCR oligomerization has been proposed to modulate trafficking, cooperativity of ligand-binding, and signaling efficacy (*Farran, 2017*). Accumulating experimental evidence suggests that certain class-A GPCRs form homo- and hetero-oligomers, in both transfected cells as well as extracted tissues. Symmetrical, parallel, dimeric interfaces have been observed in multiple crystal structures of class-A GPCRs, supporting the biological relevance of GPCR dimerization (*Huang et al., 2013*; *Manglik et al., 2012*; *Huang et al., 2015*; *Yoshikawa et al., 2012*; *Liu et al., 2012*). However, non-symmetric dimerization is rarely seen in crystal structures of class-A GPCRs. A case similar to what we observed in DRD4 crystal is revealed in a previously reported complex structure of the $\alpha_{2A}$ receptor crystallized with a high-affinity antagonist (*Liu et al., 2012*). Nevertheless, in the $\alpha_{2A}$ receptor case lipids and cholesterols were trapped in the interface groove, preventing any direct contact between the protomers. Moreover, our structural observations are in agreement with results from single-molecule dynamic studies showing that dimerization and/or oligomerization of GPCRs are more dynamic than previously believed (*Scarselli et al., 2016*). More intriguingly, it had been reported that, when co-expressed, two different epitope-tagged D4 receptors can be co-immunoprecipitated, suggesting that multiple DRD4 molecules form dimeric/oligomeric complexes within the cell membrane (*Oak et al., 2000*). The observed unusual non-symmetric dimerization in our crystal structure may suggest a mechanism of concentration-dependent regulation of the DRD4 activity in membrane, which deserves more rigorous in vivo verification in the future.

## Materials and methods

### Key resources table

| Reagent type (species) or resource | Designation | Source or reference | Identifiers | Additional information |
|---|---|---|---|---|
| Gene (*Mouse*) | DRD4 | SUNGENE BIOTECH | GenBank: BC051421.1 | |
| Cell line (*Spodoptera frugiperda*) | Sf9 and High5 | Thermo Fisher Scientific | Sf9: B825-01 High5:B85502 | |
| Cell line (*Homo sapiens*) | HEK293t | ATCC | catalog numbers: CRL-3216 | |
| Transfected construct (*Spodoptera frugiperda*) | pFastBac1-DRD4-BRIL | This paper | | Crystal construct |
| Transfected construct (*Homo sapiens*) | pcDNA3.1-DRD4-cpGFP | This paper | | Fluorescent sensor-based ligand-binding assay |

*Continued on next page*

*Continued*

| Reagent type (species) or resource | Designation | Source or reference | Identifiers | Additional information |
|---|---|---|---|---|
| Recombinant DNA reagent | pFastBac1 | Thermo Fisher Scientific | catalog numbers: 10359016 | |
| Recombinant DNA reagent | pcDNA3.1 (plasmid) | Thermo Fisher Scientific | catalog numbers: V79020 | Mammalian expression vector |
| Commercial assay or kit | Bac-to-Bac Baculovirus Expression system | Thermo Fisher Scientific | Catalog number: 10359016 | |
| Chemical compound, drug | L745870 | Tocris | Catalog number: 158985-00-3 | |
| Software, algorithm | XDS program package | *Kabsch, 2010* | | |
| Software, algorithm | PHENIX | *Adams et al., 2010* | | |
| Software, algorithm | Maestro | SCHRÖDINGER | | |
| Software, algorithm | Fiji | *Schindelin et al., 2012* | | |

## Expression and purification of DRD4-BRIL

We generated a mouse DRD4 construct containing the A1–L106 region of the BRIL (with the point mutations M7W, H102I, and R106L) from *Escherichia coli* in the place of residues A220$^{5.70}$–A302$^{6.24}$ of the receptor ICL3 (*Chun et al., 2012*). This chimera construct of DRD4 was subcloned into a modified pFastBac1 vector (Thermo Fisher) that contained an expression cassette with a hemagglutinin (HA) signal sequence followed by a FLAG-tag at the N-terminus of the receptor sequence and a PreScission Protease (PPase) recognition site followed by EGFP (enhanced GFP) and a His$_{10}$-tag at the C-terminus. The fusion protein was expressed using the Bac-to-Bac Baculovirus Expression System (Thermo Fisher) in High 5 strain of *Spodoptera frugiperda* cells for 48 hr. Harvested cells were then flash-frozen in liquid nitrogen and stored in –80°C until further use.

Cell membranes were obtained by Dounce homogenization using purification buffer A [25 mM HEPES (pH 7.4) and 150 mM NaCl]. Washed membranes were resuspended in buffer B [25 mM HEPES (pH 7.4), 150 mM NaCl, 30 μM L745870 (Tocris, Cat.# 158985-00-3), and 0.2% (w/v) iodoacetamide], with 80 mL buffer B per 0.4 L cultured biomass, and incubated at 4°C for 1 hr. The membrane was then solubilized by adding a detergent mixture [final 1% (w/v) n-dodecyl-β-D-maltopyranoside (DDM; Anatrace), and 0.15% (w/v) cholesteryl hemisuccinate (CHS; Sigma Aldrich)] for 1.5 hr at 4°C. The supernatant was isolated by centrifugation at 150,000 × *g* for 30 min. TALON IMAC resin (2 mL; Clontech) in 5 mM imidazole (pH 7.5) was added to the supernatant. The mixture was incubated at 4°C for 2–3 hr. The resin was transferred into a gravity column and washed with 15 column volumes of wash buffer C [25 mM HEPES (pH 7.4), 150 mM NaCl, 0.05% (w/v) DDM, 0.015% (w/v) CHS, 30 mM imidazole, and 30 μM L745870]. The resin-binding protein was then treated overnight with homemade His-tagged PPase in 10 mL buffer C. The flow through was then concentrated to 1 mL before being loaded onto a Superdex-200 column (GE healthcare). Fractions were collected from the monodispersed peak (*Figure 1—figure supplement 1b*) and pooled into one tube. L745870 (final concentration 50 μM) was added to the pooled protein sample and equilibrated for 1 hr before being concentrated to ~60 mg/mL with a 50 kDa cut-off centrifuge concentrator (Millipore).

## Lipidic cubic phase crystallization

The protein sample was reconstituted into LCP by mixing 40% sample with 60% lipid [10% (w/w) cholesterol, and 90% (w/w) monoolein] using the twin-syringe method (*Caffrey, 2009*). The protein-lipid mixture was dispensed as 40 nL drops onto glass sandwich plates (Shanghai FAstal BioTechô Inc) using a hand-held dispenser (Art Robbins Instruments) and overlaid with 0.8 μL of precipitant solution. Protein crystals were obtained from precipitant solution containing 100 mM citrate acid/NaOH (pH 5.0–6.0), 0–150 mM salts, and 30% PEG550DME or PEG600 (*Figure 1*-figure Supplements 1c$_{i-iv}$). The best crystals grew in 100 mM MES (pH 6.0), 50 mM ammonium citrate, and 30% PEG400 (*Figure 1—figure supplement 1cv*). Crystals grew to a full size of ~50 μm×20 μm×5 μm

within one week, and were harvested directly from the LCP matrix using 50 µm micromounts (MiTe-Gen, Cat.# M2-L19-50), before flash-frozen in liquid nitrogen until further use.

## Data collection, structure solution and refinement

X-ray diffraction data were collected at the SPring-8 beamline 41XU, Hyogo, Japan, using an EIGER16M detector (Dectris; X-ray wavelength 1.0000 Å). A raster system was used to locate the best diffracting parts of single crystals (*Cherezov et al., 2009*). The crystals were exposed to a 10 µm unattenuated mini-beam for 0.1 s and 0.2° oscillation per frame. Most crystals of the DRD4-L745870 complex diffracted to a resolution of 3.5–4.0 Å. The software package, XDS (*Kabsch, 2010*), was used for integrating and scaling data from 28 best-diffracting crystals.

The DRD4-L74570 complex was solved by molecular replacement using the PHENIX program (*Adams et al., 2010*; *Emsley et al., 2010*) and two independent search models, namely, the receptor portion of human DRD4 (PDB ID: 5WIU) (*Wang et al., 2017*) and the BRIL protein (PDB ID: 1M6T) (*Chu et al., 2002*). Refinement was performed with PHENIX_REFINE, iterated with visual examination and rebuilding of the refined model using the program COOT (*Emsley et al., 2010*) and $|2F_o|-|F_c|$, $|F_o|-|F_c|$, and omit maps. Simulation of ligand docking in receptor was done with Maestro (SCHRÖDINGER)(*Friesner et al., 2006*).

## Fluorescent sensor-based ligand-binding assay

Sequences encoding a cleavable HA secretion-signaling motif and a FLAG epitope were placed at the 5'-end of the DRD4 constructs (WT* and 'Crystal' in *Figure 2—figure supplement 1a*). *Sal*I and *Not*I cleavage sites were created at the 5'- and 3'-ends, respectively, for cloning into pcDNA-3.1 (Thermo Fisher) to generate all mammalian cell-expression constructs.

Coupling between conformational changes and chromophore fluorescence were anticipated to follow the reported example of dLight1 (*Patriarchi et al., 2018*) which is a DRD2 variant. In particular, the cpGFP module (LSSLE-cpGFP-LPDQL) was copied from the plasmid GCaMP6 (*Chen et al., 2013*) and inserted into DRD4 variants of WT* and point mutations as well as the 'crystallization' construct to replace the segment of R217$^{5.67}$–W306$^{6.28}$.

## Cell culture, imaging, and quantification

HEK293T cells were cultured in glass-bottom cell-culture dish (NEST, Cat.# 801001) and transfected with DRD4-cpGFP expression plasmid using polyetherimide (Sigma Aldrich). Cell imaging was performed using a 60 × oil based objective on an inverted OLYMPUS FV1200 laser scanning confocal microscope with 488/513 nm ex/em wavelengths. For testing DRD4-cpGFP responses, different ligands were directly applied to the bath during the imaging session. Reproducible results were obtained after repeating all assays 2–3 times. Mean intensity of ROI (region of interested) in every frame was calculated using the Fiji software (*Schindelin et al., 2012*). No commonly misidentified cell lines were used. The cell lines are negative for mycoplasma contamination.

## Protein cross-linking

Purified protein (1 mg/mL) was incubated with serial dilutions (0–5 mM) of glutaraldehyde dissolved in reaction buffer [25 mM HEPES (pH 7.4), 150 mM NaCl, 0.05% (w/v) DDM, and 0.015% (w/v) CHS] at room temperature for 1 hr. Cross-linking was stopped by adding in 5 × loading buffer [250 mM Tris-HCl (pH 6.8), 10% (w/v) SDS, 0.5% (w/v) bromophenol blue, 50% (v/v) glycerol, and 5% (w/v) β-Mercaptoethanol] before the sample was analyzed by SDS-PAGE.

## Acknowledgements

The authors thank staffs of the Protein Science Core Facility of the Institute of Biophysics for their excellent assistance in protein characterization and thank Ms. Yan Teng of the Biological Imaging Core Facility for helping in confocal microscopy analysis. The synchrotron radiation experiments were performed at BL41XU of SPring-8 with approval from the Japan Synchrotron Radiation Research Institute (proposal number: 2018A2559). We thank the BL41XU beamline staff members for help with X-ray data collection. We thank Ms. Bei Yang for experimental assistance. We are particularly thankful to Dr. Yulong Li of the Peking University for the kind gift of the plasmid of the

DRD2-cpGFP fusion construct. We also thank Dr. Torsten Juelich (Peking University, China) for linguistic assistance during the preparation of this manuscript. This work was supported by the CAS Strategic Priority Research Program (XDB08020301), the Ministry of Science and Technology (China) (2015CB910104), and the National Natural Science Foundation of China (31470745).

## Additional information

### Funding

| Funder | Grant reference number | Author |
|---|---|---|
| Chinese Academy of Sciences | Strategic Priority Research Program XDB08020301 | Xuejun Cai Zhang |
| Ministry of Science and Technology | 2015CB910104 | Xuejun Cai Zhang |
| National Natural Science Foundation of China | 31470745 | Xuejun Cai Zhang |

The funders had no role in study design, data collection and interpretation, or the decision to submit the work for publication.

### Author contributions

Ye Zhou, Conceptualization, Resources, Data curation, Formal analysis, Validation, Investigation, Visualization, Methodology, Writing—original draft, Writing—review and editing; Can Cao, Conceptualization, Data curation, Validation, Methodology; Lingli He, Data curation, Software, Methodology; Xianping Wang, Conceptualization, Resources, Supervision, Funding acquisition, Project administration; Xuejun Cai Zhang, Conceptualization, Supervision, Funding acquisition, Writing—original draft, Project administration, Writing—review and editing

### Author ORCIDs

Ye Zhou [ID] https://orcid.org/0000-0002-0489-3614
Xuejun Cai Zhang [ID] https://orcid.org/0000-0001-6726-3698

### Decision letter and Author response

Decision letter https://doi.org/10.7554/eLife.48822.sa1
Author response https://doi.org/10.7554/eLife.48822.sa2

## Additional files

### Supplementary files

• Supplementary file 1. Data collection and refinement statistics for the mouse DRD4 and L745870 complex.[a]. Values in parentheses present the highest resolution shell. $CC_{1/2}$** (*Diederichs and Karplus, 2013*). [b]. All outliers are located in loops not involved in ligand binding.
• Transparent reporting form

### Data availability

Diffraction data have been deposited in PDB under the accession code 6IQL.

The following dataset was generated:

| Author(s) | Year | Dataset title | Dataset URL | Database and Identifier |
|---|---|---|---|---|
| Ye Z, Can Cao, Xuejun Cai Zhang | 2019 | Crystal structure of dopamine receptor D4 bound to the subtype-selective ligand, L745870 | http://www.rcsb.org/structure/6IQL | Protein Data Bank, 6IQL |

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
