## [Decision Letter]

**Acceptance summary:**

This manuscript described the crystal structure of mouse dopamine D4 in complex with a selective antagonist L745870. The structure, together with earlier human D4 structure, revealed an extended binding pocket in addition to the orthosteric pocket. The structure also explained the subtype selectivity of L745870 as this ligand inserted into the extended binding pocket formed by helices II and III in D4, in contrast to a different extended pocket in D2/D3. Using the cpGFP-fusion assay combined with mutagenesis study, the author proposed a reasonable antagonism mechanism of L745870. A unique feature is that they found the structure is a non-symmetric dimer in crystal and tried to verify it in solution. While there are still debates about the dimeric conformation and necessity of GPCR, but it is surely meaningful to see and discuss this non-symmetric conformation. In summary, this paper reveals the ligand selectivity among different dopamine receptors and also shows how L745870 series ligands can inhibit receptor activity.

**Decision letter after peer review:**

Thank you for submitting your article "Crystal structure of dopamine receptor D4 bound to the subtype-selective ligand, L745870: selectivity and antagonism" for consideration by *eLife*. Your article has been reviewed by two peer reviewers, and the evaluation has been overseen by a Reviewing Editor and Olga Boudker as the Senior Editor. The following individual involved in review of your submission has agreed to reveal their identity: Zhi-Jie Liu (Reviewer #1).

The reviewers have discussed the reviews with one another and the Reviewing Editor has drafted this decision to help you prepare a revised submission.

Essential revisions:

1) The cross-linking study should be conducted while proteins are on the cell membrane as this will mimic and trap the physiological non-symmetry dimer; in contrast, proteins in detergent micelle may be linked by glutaraldehyde artificially. Alternatively, the author may add a monomeric GPCR as negative control with the current method. What are the cross-linkable residues on the identified dimer interface that may be responsible for Figure 3—figure supplement 1?

2) Is there a pharmacological study comparing the efficiency of ligand L745870 against mouse and human D4 receptor? Can we get some clue based on the current crystal structure and sequence alignment between human and mouse? Since the human D4 structure is also available is it possible to discuss how ligand potency for either L745870 or other ligands can be improved?

3) Based on Figure 1—figure supplement 1, L745870 appears to have lost significant potency on the engineered constructs. The results in that figure are difficult to interpret, as "L-7" was always in micromolar range. Some routine molecular pharmacology, such as radioligand binding or functional assays would help to convincingly confirm that the crystallization construct retains high affinity for L745870.

4) The structure is of relatively low resolution (3.5 Å), how did the authors determine and confirm the orientation of L745870, which has a linear shape, in the binding site? Some simple molecular dynamics simulations showing the stability of the binding pose would strengthen the argument, especially given that Figure 1C shows L745870 may have different binding poses in Mol-A and Mol-B.

5) While the analysis recapitulates previously identified TM2 and TM3 micro-patch on receptor side, it is not clear why L745870 has much higher affinity for DRD4 than for DRD3 and DRD2 on the ligand side, or why L745870 is selective but nemonapride is not, as nemonapride protrudes into the EBP shown in Figure 2E as well. Either more discussion or more analysis is warranted to further elaborate this point.

---

## [Author Response]

Essential revisions:1) The cross-linking study should be conducted while proteins are on the cell membrane as this will mimic and trap the physiological non-symmetry dimer; in contrast, proteins in detergent micelle may be linked by glutaraldehyde artificially. Alternatively, the author may add a monomeric GPCR as negative control with the current method. What are the cross-linkable residues on the identified dimer interface that may be responsible for Figure 3—figure supplement 1?

We thank the reviewers’ suggestion. It would be more solid if we could confirm the dimer state in the in vivo environment. In fact, we did try to perform the crosslinking experiment in the cell membrane. However, the results were less clean than just in the presence of detergent; higher order aggregation and oligomerization existed after trying different concentrations of crosslinking agents. Based on our crystal structure, the most probable crosslinking sites in the detergent environment include Arg376 on Helix 8 of protomer A, and Arg216, Arg215 on the cytosol side of TM5 from protomer B.

2) Is there a pharmacological study comparing the efficiency of ligand L745870 against mouse and human D4 receptor? Can we get some clue based on the current crystal structure and sequence alignment between human and mouse? Since the human D4 structure is also available is it possible to discuss how ligand potency for either L745870 or other ligands can be improved?

A previous study (Patel et al., 1997) showed that there was no significant species difference in terms of affinity of L-745,870 towards the human and rodent D4 receptors, which is consistent with the sequence conservation and structure alignment. Meanwhile, most of the pharmaceutical evaluation of L-745,870 was performed in rodents. As one of the ligands of highest selective affinity towards D4 receptors and behaving well in pharmacokinetic trials, L-745870 remains to be a potential drug for clinic use though it failed in treating schizophrenia. New clinic trials are conducted for other disease recently. The drug design question is addressed together with “Question 5”.

3) Based on Figure 1—figure supplement 1, L745870 appears to have lost significant potency on the engineered constructs. The results in that figure are difficult to interpret, as "L-7" was always in micromolar range. Some routine molecular pharmacology, such as radioligand binding or functional assays would help to convincingly confirm that the crystallization construct retains high affinity for L745870.

Compared with radio-ligand binding, the fluorescence assay showed lower signal noise ratio and was used mainly for large scale qualitative analyses. In this assay, fluorescence signals came from dopamine binding towards the engineered GFP-D4 receptor, and the antagonists were added afterwards to the same basin. Due to the limited fluorescing window of the GFP component, the change of signal should be recorded in a short time. Therefore, relatively high concentrations of competitive ligands were used. However, the binding affinity of L745870 towards the engineered receptors likely remains high, since L745870 was able to rapidly replace dopamine in the fluorescence assay, and was required (at a concentration higher than 50 nM) for successful crystallization.

4) The structure is of relatively low resolution (3.5 Å), how did the authors determine and confirm the orientation of L745870, which has a linear shape, in the binding site? Some simple molecular dynamics simulations showing the stability of the binding pose would strengthen the argument, especially given that Figure 1C shows L745870 may have different binding poses in Mol-A and Mol-B.

Thanks the reviewers for this good suggestion. Indeed, L745870 in its free form may assume multiple stereo-chemical conformations, as being confirmed by the programs PHENIX (Liebschner et al., 2019) and Schrodinger (Friesner et al., 2006). However, Asp112 of the receptor is most likely to form a hydrogen bond with proton-titratable amine group in L745870 during the binding, which strongly restricts the freedom of ligand docking. Accordingly, previous analyses (Kortagere et al., 2004; Thal et al., 2016) have proposed a binding mode similar to our refined model. Nevertheless, we are aware that it is possible that the proton-titratable amine group in L745870 occurs in either the 1*H*-pyrrolo[2,3-*b*]pyridine side or chlorophenyl side with micro-environment changing. The difference of ligand density between the two protomers appears to be consistent with this possibility.

5) While the analysis recapitulates previously identified TM2 and TM3 micro-patch on receptor side, it is not clear why L745870 has much higher affinity for DRD4 than for DRD3 and DRD2 on the ligand side, or why L745870 is selective but nemonapride is not, as nemonapride protrudes into the EBP shown in Figure 2E as well. Either more discussion or more analysis is warranted to further elaborate this point.

We thank the reviewers for raising this interesting question. Further discussion is added to the related part in Discussion (second paragraph).

Reference:

Liebschner, D., et al., Macromolecular structure determination using X-rays, neutrons and electrons: recent developments in Phenix. Acta Crystallogr D Struct Biol, 2019. 75(Pt 10): p. 861-877.